# Transcriptome Profiling of Circulating Tumor Cells to Predict Clinical Outcomes in Metastatic Castration-Resistant Prostate Cancer

**DOI:** 10.3390/ijms24109002

**Published:** 2023-05-19

**Authors:** Levi Groen, Iris Kloots, David Englert, Kelly Seto, Lana Estafanos, Paul Smith, Gerald W. Verhaegh, Niven Mehra, Jack A. Schalken

**Affiliations:** 1Department of Urology, Radboud Institute for Molecular Life Sciences, Radboud University Medical Center, 6525 GA Nijmegen, The Netherlands; levi.groen@radboudumc.nl (L.G.); gerald.verhaegh@radboudumc.nl (G.W.V.); 2Department of Medical Oncology, Radboud University Medical Center, 6525 GA Nijmegen, The Netherlands; iris.kloots@radboudumc.nl (I.K.); niven.mehra@radboudumc.nl (N.M.); 3ANGLE Biosciences Inc., Toronto, ON M9W 1B3, Canada; d.englert@rnasence.com (D.E.); k.seto@rnasence.com (K.S.); l.estafanos@rnasence.com (L.E.); p.smith@rnasence.com (P.S.)

**Keywords:** prostate cancer, castration resistant prostate cancer, circulating tumor cells, transcriptomics, prognostic, predictive, liquid biopsy, biomarkers, androgen receptor signaling inhibitors, chemotherapy

## Abstract

The clinical utility of circulating tumor cells (CTC) as a non-invasive multipurpose biomarker is broadly recognized. The earliest methods for enriching CTCs from whole blood rely on antibody-based positive selection. The prognostic utility of CTC enumeration using positive selection with the FDA-approved CellSearch^TM^ system has been demonstrated in numerous studies. The capture of cells with specific protein phenotypes does not fully represent cancer heterogeneity and therefore does not realize the prognostic potential of CTC liquid biopsies. To avoid this selection bias, CTC enrichment based on size and deformability may provide better fidelity, i.e., facilitate the characterization of CTCs with any phenotype. In this study, the recently FDA-approved Parsortix^®^ technology was used to enrich CTCs from prostate cancer (PCa) patients for transcriptome analysis using HyCEAD^TM^ technology. A tailored PCa gene panel allowed us to stratify metastatic castration-resistant prostate cancer (mCRPC) patients with clinical outcomes. In addition, our findings suggest that targeted CTC transcriptome profiling may be predictive of therapy response.

## 1. Introduction

The discovery of oncogenes in the 1970s led to the development of targeted therapies and initiated new hope for curative cancer treatments [1]. Solid tumor genotyping has revealed novel druggable oncogenic alterations that contribute to therapy resistance in patients with mCRPC. This has led to the development of novel treatment modalities such as poly ADP-ribose polymerase inhibitors (PARPi) for tumors with multiple loss-of-function alterations in DNA-repair genes [2]. Unfortunately, response rates have been highly variable and unpredictable [3].

Continued reliance on solid tissue biopsies to identify druggable alterations may be limiting progress. Conventional biopsies do not capture clonal heterogeneity and are limited by their invasive nature and the inaccessibility of certain lesions [4]. Inadequate insight into dynamic spatiotemporal alterations in PCa tumors has restricted the overall clinical benefit of novel therapies. Furthermore, the detection limit of traditional diagnostic imaging techniques is currently about one billion cells; below this threshold, malignant lesions are undetectable [5]. Together, these lesions can present a significant tumor burden, yield a high degree of heterogeneity, and carry underlying therapy resistance-associated genetic aberrations [6].

Though CellSearch was introduced in 2005, CTC enumeration has not been implemented as a prognostic, or an early response biomarker in daily clinical PCa management [7]. Recent developments in molecular profiling and cell sorting technologies have increased sensitivity and decreased costs. This could make CTC genotyping viable for precision diagnostics in the near future [8,9]. The collection of tumor material via liquid biopsies allows oncologists to deliver precision medicine and stratify therapy response by the detection of biomarkers, such as *androgen receptor splice variant 7* (*ARv7*), in CTCs [10,11]. With longitudinal sampling, CTC transcriptomes present temporally accurate tumor phenotypes with clinically actionable potential [12]. While biomarker discovery is a growing field in clinical research, therapeutic innovations have outpaced diagnostic capabilities in oncology. As a result, patient stratification for personalized therapeutics remains an unmet clinical need.

In the present work, we used label-free CTC enrichment coupled with CTC transcriptome profiling to characterize multiple oncogenic pathways in mCRPC patients starting a new line of therapy. The Parsortix PR1 microfluidic system, which utilizes the same technology as the FDA-approved PC1 system, was used to enrich CTCs from whole blood using physical properties such as a larger size and reduced deformability compared to leukocytes [13]. This approach enables the capture of epithelial and mesenchymal CTCs in both single and aggregate forms (CTC clusters) [13]. The resulting tumor-enriched samples contain CTCs in a background of 200–800 leukocytes per mL of processed blood [13]. CTC-enriched samples were analyzed using a patented hybridization technology known as HyCEAD [14,15]. This multiplex gene expression assay provides the required sensitivity and specificity to enable the detection of a single CTC target in a background of other circulating epithelial cells and leukocytes [15]. HyCEAD has previously been used to predict malignancy in women with pelvic masses [16].

The primary aim of this study was to test the feasibility of label-free CTC enrichment and targeted transcriptome profiling in mCRPC to prognosticate patients. A secondary aim was identifying transcriptome profiles predictive of therapy response in CTC-enriched mCRPC samples.

## 2. Results

### 2.1. CTC Transcriptome Gene Panel

The CTC transcriptome panel consists of genes with high expression in PCa and low expression in leukocytes. Genes were selected as described in Section 4.1. In total, 64 genes were included; each gene and its relevance to PCa are shown in Appendix A.

### 2.2. Clinical Follow-Up

In total, 40 patients provided baseline CTC samples (i.e., before starting a new line of therapy). Patients received various therapies: androgen receptor inhibitors (ARSI) (abiraterone or enzalutamide; *n* = 22), chemotherapy (docetaxel or cabazitaxel; *n* = 9), immunotherapy (Ipilimumab and Nivolumab; *n* = 2), radioligand therapy (PSMA-Lu177 or Radium-223; *n* = 2), PARPi (Olaparib; *n* = 2), and two were actively surveilled until signs of disease progression (i.e., they did not receive systemic therapy). The clinical trajectories, time of inclusion (CTC collection), and lines of therapy are shown in Figure 1. In one case, two CTC samples were collected from a patient receiving PARPi, ‘RAD30’ at baseline followed by ‘RAD41’ 14 days later (Appendix A). ‘RAD41’ was excluded from the primary analysis as it was not collected at baseline. Baseline alkaline phosphatase and PSA levels, age, de novo metastasis, years since CRPC diagnosis, type of therapy received following CTC collection, and 1-year survival per therapy group are shown in Table 1.

### 2.3. Genomic Alterations and CTC Transcriptomes

Next-generation DNA sequencing of solid tissue biopsies was done successfully in 38/40 patients. *AR* alterations were detected in 11 patients (29%): nine amplifications, and three mutations. In one patient, *AR* was amplified and mutated. *TP53* alterations were found in 10 patients (26%): nine mutations, and one shallow deletion. *PTEN* alterations were identified in seven patients (18%): four mutations, and three shallow deletions. In nine patients (24%), there were alterations in DNA damage repair (DDR) genes: *BRCA2*, *MSH2*, *RAD51B*, *CHEK2*, and *FANCL*. *BRCA2* was most frequently altered with three mutations and two shallow deletions. Mutations in *PPP2R2A*, *CDK12*, *MSH2*, *RAD51B*, *CHEK2*, *RB1*, *PIK3CA*, *AKT1*, *SPOP*, and *FANCL* were present at lower frequencies (2.6–5%) (Appendix A).

Due to overlap in the DNA sequencing and CTC transcriptome panels, *AR* and *BRCA2* genomic alterations in solid tissue biopsies were associated with expression in CTCs. In eight of nine patients with *AR* amplifications, the expression of *AR* and/or associated signaling genes (*KLK3*, *ARv7*, and *FOXA1*) was increased in matched CTC samples, suggesting high concordance between solid tissue and liquid CTC biopsies [17]. In the patient with both *AR* amplification and mutation, expression of *AR*, *KLK3*, *ARv7*, and/or *FOXA1* were not increased in CTCs (Appendix A). Surprisingly, in two patients with *BRCA2* shallow deletions, *BRCA2* and associated DDR genes (*BRCA1*, *FANCA*, *RRM2*, and *TOP2A*) were expressed in matched CTCs in one case (Appendix A). This suggests the loss of *BRCA2* may not always be concordant between solid tissue and CTCs biopsies [18].

### 2.4. Prognostic Value of CTC Profiling

Hierarchical clustering of patients irrespective of treatment modality (referred to as the ‘therapy-agnostic cohort’) with the complete gene panel yielded two transcriptionally distinct groups (Appendix A). Patients with high expression of AR signaling genes (*ARv7*, *DLX1*, *HOXB13*, and *KLK3*), DDR genes (*BRCA1*, *BRCA2*, FANCA, and *TOP2A*), and oncogenes (*ERG* and *GRHL2*) had reduced progression-free survival (PFS) (hazard ratio (HR) = 1.99, *p* = 0.076), and significantly reduced overall survival (OS) (HR = 5.1, *p* = 0.007) (Appendix A). Genes with limited prognostic value were excluded from subsequent clustering to improve patient prognostication.

The relative prognostic value of genes from the complete gene panel was determined via multivariate analysis on PFS. Genes were ranked using HR (Appendix A). Notably, the epithelial marker, *EPCAM*, was not a high-ranking prognosticator in this cohort, yet CTCs expressing *EPCAM* are known to be prognostically relevant in various malignancies [7,19]. Instead, genes involved in AR signaling (*KLK3*, *HOXB13,* and *GRHL2*), and metastasis (*MYO6* and *TRPM8*) were more informative [20,21]. Prognostic candidates were cross-referenced with relevant literature and went through several iterations of clustering/survival analyses to form a tailored panel consisting of the following genes: *AR*, *ARv7*, *FOLH1* (aka PSMA), *KLK2*, *KLK3*, and *TMPRSS2* (hereafter referred to as the ‘agnostic gene panel’) [22].

CTC profiling using the agnostic gene panel produced two transcriptionally distinct groups, ‘Group 1’ and Group 2’ (Figure 2A). There were no significant differences in *PTPRC* levels between these groups (Appendix A). At baseline, ‘Group 1’ patients (*n* = 16) had lower mean leukocyte counts (6.7 vs. 9.2 × 10^9^/L), and higher mean levels of alkaline phosphatase (334.5 vs. 103.6 U/L) and PSA (215.8 vs. 40.8 µg/L) compared to ‘Group 2’ patients (*n* = 24) (Appendix A). Risk for progression was significantly increased for patients in ‘Group 1’ (HR = 4.28, *p* < 0.001) (Figure 2B). In a univariate model, the ‘Group 1’ profile was the best prognosticator for PFS (HR = 4.28, *p* < 0.001) compared to PSA (HR = 1.64, *p* = 0.214), ARv7 (HR = 1.26, *p* = 0.551), and age (HR = 0.99, *p* = 0.971) (Figure 2D). The ‘Group 1’ profile remained the strongest prognosticator for PFS in a multivariate model (HR = 3.83, *p* = 0.002) containing age (HR = 1.00, *p* = 0.850) and PSA (HR = 1.00, *p* = 0.045) (Appendix A). ‘Group 1’ patients also had a significantly increased risk for overall mortality (HR = 10.78, *p* < 0.001), where 63% of patients in ‘Group 1’ were deceased versus 8% of those in ‘Group 2’ at last follow-up (Figure 2C and Appendix A). In a univariate model, the ‘Group 1’ profile was a better prognosticator for OS (HR = 10.78, *p* = 0.002) than age (HR = 2.44, *p* = 0.146), *ARv7* expression (HR = 2.33, *p* = 0.170), and PSA (HR = 1.79, *p* = 0.321) (Figure 2E and Appendix A). Likewise, the ‘Group 1’ profile was highly prognostic for OS (HR = 9.99, *p* = 0.005) in a multivariate model containing age (HR = 1.04, *p* = 0.199) and PSA (HR = 1.00, *p* = 0.214) (Appendix A).

### 2.5. CTC Profiles Predict Therapy Response

To explore whether targeted CTC transcriptome profiling has predictive value regarding therapy responses, patients were assessed based on their subsequent therapy.

Genes with the potential to predict an ARSI response were identified via a multivariate analysis on PFS using the subset of ARSI patients (*n* = 22) and the complete gene panel. Genes were ranked using HR (Appendix A). Amongst the top-ranking genes, most were involved in AR signaling such as *GRHL2*, *KLK3*, *HOXB13*, *FOXA1*, and *AGR2*. Several iterations of clustering and survival analyses produced a gene panel (aka the ‘ARSI gene panel’) with improved prognostic value over the complete gene panel (Figure 3A–C and Appendix A).

Hierarchical clustering using the ARSI gene panel produced two transcriptionally distinct groups referred to as ‘ARSI 1’ and ‘ARSI 2’ (Figure 3A). There were no significant differences in *PTPRC* levels between these groups (Appendix A). At baseline, ‘ARSI 1’ patients (*n* = 6) had lower mean leukocyte counts (7 vs. 10.3 × 10^9^/L), and higher mean levels of alkaline phosphatase (255.3 vs. 99.9 U/L) and PSA (130.1 vs. 39.8 µg/L) compared to ‘ARSI 2’ patients (*n* = 16) (Appendix A). Risk for progression was significantly increased for patients in ‘ARSI 1’ (HR = 13.05, *p* < 0.001) (Figure 3B). Likewise, ‘ARSI 1’ patients had an increased risk for overall mortality (HR = 21.56, *p* < 0.001), where 88% of ‘ARSI 1’ patients were deceased versus 6% of ‘ARSI 2’ patients at last follow-up (Figure 3C and Appendix A). In a univariate model, the ‘ARSI 1’ profile was the best prognosticator for PFS (HR = 13.05, *p* = 0.002) when compared to *ARv7* expression (HR = 1.76, *p* = 0.282), PSA (HR = 1.35, *p* = 0.566), and age (HR = 0.78, *p* = 0.636) (Figure 3D). This was validated in a multivariate model containing age and PSA, where the ‘ARSI 1’ profile carried a significantly increased risk for progression (HR = 11.59, *p* = 0.006) (Appendix A). Interestingly, the prognostic value of *ARv7* expression on OS was higher in the ARSI cohort than the therapy-agnostic cohort (HR = 6.90 vs. 2.33) (Figure 2E, Figure 3E and Appendix A). This was not the case for PFS (Figure 2D and Figure 3D). In a univariate model, ‘ARSI 1’ was the strongest prognosticator for OS (HR = 21.56, *p* = 0.005) when compared to *ARv7* expression (HR = 6.90, *p* = 0.080), PSA (HR = 2.53, *p* = 0.291), and age (HR = 1.76, *p* = 0.518) (Figure 3E). This was validated in a multivariate model including age and PSA, where the ‘ARSI 1’ profile was the strongest prognosticator for OS (HR = 27.36, *p* = 0.005) (Appendix A).

To identify genes with predictive potential for chemotherapy response, the complete gene panel was analyzed in a multivariate model on PFS using the subset of chemotherapy patients (*n* = 9). Genes were ranked using HR (Appendix A). *AKR1C3*, *DLX1*, *HOXC6*, and *MYO6* were among the top-ranked genes. Kaplan–Meier curves on PFS and OS for *ARv7* expression, age, PSA, and the above-mentioned genes can be seen in Figure 4 and Figure 5. Expression of *AKR1C3*, *DLX1*, *HOXC6*, and *MYO6* were equally strong prognosticators for PFS (HR = 10.97, *p* = 0.011) and ranked above age (HR = 0.8, *p* = 0.806), *ARv7* (HR = 2.45, *p* = 0.320), and PSA (HR = 0.91, *p* = 0.923) (Figure 4A–G). Interestingly, the prognostic value of *ARv7* expression on OS ranked above all other considered variables in this cohort (HR = 6.82, *p* = 0.081) (Figure 5A–G). Due to the small sample size and the accompanied statistical limitations, no further analysis was done in this cohort.

### 2.6. Longitudinal CTC Profiling—A Case Discussion

Divergent from the main protocol, one patient was sampled twice: a baseline sample, ‘RAD30’, and a follow-up sample, ‘RAD41’, 14 days after starting PARPi therapy. Both samples had comparable *PTPRC* levels and relatively similar CTC transcriptome profiles, with some notable exceptions. A clear upregulation of *SOX2*, *TRPM8*, *NAALADL2*, *ARv7*, *DLX1*, *TUSC3*, *GHR*, *FOXA1*, *FOLH1*, and *HOXC6* along with a downregulation of *WNT5A* expression can be seen in ‘RAD41’ when compared to baseline. A smaller, but noticeable, increase in *EPCAM* expression indicates the epithelial cell fraction in ‘RAD41’ may be higher than in ‘RAD30’ (Appendix A). Many of the upregulated genes are associated with AR signaling, suggesting an increase in AR pathway activity in ‘RAD41’, either via changes in gene expression or CTC load.

## 3. Discussion

The present work highlights the prognostic utility of CTC transcriptome profiling in advanced PCa. Based on transcriptome analysis of tumor tissue samples, 64 genes were selected for targeted CTC transcriptome profiling. From this gene panel, AR signaling genes were amongst the strongest prognosticators, yielding two transcriptionally distinct groups with differing clinical trajectories. A recent publication from Sperger et al. corroborates these findings [22]. While the present study focused on advanced PCa, the Sperger study included a significant portion of castration-sensitive prostate cancer (CSPC) patients (28%). As highlighted by the considerable overlap between the Sperger-gene panel and our therapy-agnostic gene panel, AR signaling continues to drive both CSPC and mCRPC progression. In recent years, *ARv7* has been a primary focus for biomarker research in PCa [11,23]. Its ability to drive AR signaling in the absence of a ligand would suggest *ARv7* to be prognostically useful in mCRPC [23]. However, we found CTC *ARv7* expression to be an insignificant independent prognosticator for survival in the therapy-agnostic cohort, likely due to the presence of alternative resistance mechanisms [24].

In patients receiving ARSI therapy, CTC *ARv7* expression did have prognostic value in terms of OS but not PFS. Not surprisingly, AR signaling genes remained strong prognosticators for survival in the ARSI cohort. Stratifying therapy response in patients receiving ARSIs using a tailored gene panel targeting AR pathway activity significantly improved the prognostic value of CTC transcriptome profiling (Figure 3A and Appendix A). Scoring AR pathway activity has been successfully used to stratify response to hormone therapy in hormone-sensitive AR-positive salivary duct carcinoma (SDC) [25]. Interestingly, *SRD5A1* expression was a stronger prognosticator than AR pathway activity in hormone-sensitive SDC [25]. *SRD5A1* increases intracellular dihydrotestosterone (DHT) indicating canonical (androgen-dependent) AR pathway activity [26]. Unlike other AR-associated genes, *SRD5A1* expression was unable to stratify the ARSI response in the present study, underscoring the prevalence of non-canonical AR signaling in these patients (Appendix A) [27]. *AR*/*ARv7* also lacked prognostic utility as independent biomarkers for PFS in the ARSI cohort. The expression of *AR*/*ARv7* alone is not informative of AR nuclear translocation, which is a prerequisite for genomic signaling [28]. Instead, the ability of *AR*/*ARv7* to prognosticate PFS relied on the co-expression of *AR* co-factors and their transcriptional target genes. These findings highlight the importance of using a multifaceted approach when profiling AR pathway activity for stratifying ARSI response in mCRPC.

CTC transcriptome profiling in patients receiving chemotherapy yields a surprisingly different set of prognostic genes when compared to those receiving ARSIs (Appendix A). *AKR1C3*, *DLX1*, *HOXC6*, and *MYO6* were the strongest prognosticators for survival in this cohort.

*DLX1* and *HOXC6* are primary biomarkers in the SelectMDx liquid biopsy test for the diagnosis of high-risk localized PCa [29]. *HOXC6* expression is upregulated in localized, advanced, and metastatic PCa, promoting proliferation [30]. As an androgen-independent AR co-factor, *HOXC6* can drive noncanonical AR signaling under castrate conditions and theoretically promote *DLX1* expression in ERG-negative mCRPC [30,31,32]. This could explain their consistent prognostic utility in both CTC and urinary liquid biopsy studies [33]. Combined, *HOXC6* and *DLX1* could act synergistically with regards to driving progression in mCRPC patients receiving chemotherapy, since upregulation of *DLX1* promotes metastasis in mice with advanced PCa [32].

*AKR1C3* is an androgenic enzyme that is highly expressed in mCRPC. Through intracellular biosynthesis of DHT, AKR1C3 can drive canonical AR signaling in the absence of testosterone [34]. Surprisingly, *AKR1C3* is also highly expressed in AR-negative mCRPC, pointing to an AR-independent oncogenic function [34]. One such moonlighting function was found in esophageal adenocarcinoma where AKR1C3 regulates AKT phosphorylation, leading to chemotherapy resistance [35].

*MYO6* encodes an actin motor protein with a key function in cell motility [36]. *MYO6* is upregulated in PCa and promotes the proliferation of CRPC cells [37]. *MYO6* may be of importance for metastasis, as the knockdown of *MYO6* mRNA in PCa cell lines impairs cellular migration [38]. In this light, CTC *MYO6* expression could be reflective of metastatic potential. We found *MYO6* to be uniquely prognostic in patients receiving chemotherapy, a cohort with a high metastatic burden as reflected by their alkaline phosphatase levels (Appendix A).

As we were unable to determine CTC counts, it is difficult to elucidate its impact on CTC transcriptome profiles. However, *EPCAM*, the cell adhesion marker for prognostic CTC-enumeration in the CellSearch platform, was not a strong prognosticator in this study (Appendix A) [7]. Thus, CTC transcriptome profiling likely carries added benefit above enumeration.

We found AR pathway activity to be a prognosticator across treatment modalities, underscoring its broad scope of oncogenic functions in mCRPC. Recent studies have shown synthetic lethality between AR and PARP inhibition, implying a functional role of AR in DDR [39,40]. PARP inhibition reduces genomic stability and inhibits AR nuclear translocation in PCa [39,40,41]. However, we found the expression of several AR signaling genes upregulated in CTCs of a patient receiving PARPi when compared to baseline (Appendix A). This patient harbored a *BRCA2* mutation, which has been associated with increased Src signaling [42]. Phosphorylation of AR by Src kinase promotes AR nuclear translocation constituting a potential mechanism for PARPi resistance [42]. Monitoring CTC AR pathway activity could serve as an early response biomarker for patients with *BRCA2* mutations receiving PARPi. Further research is needed to substantiate this claim.

So far, this discussion has focused on the clinical utility of CTC transcriptome profiling in mCRPC. However, CTC survival, invasiveness, and ability to resist vascular stressors are unlikely to be mCRPC-specific [43,44]. CTC clusters are associated with worse clinical outcomes in several cancers due to their resilience to vascular stressors and high metastatic potential [44]. The formation of CTC clusters is highly dependent on the expression of the basal cell marker, *KRT14*, in ovarian and breast cancer [44,45]. Basal PCa cells are known for their aggressive phenotype [46]. The cell membrane channel, TRPM8, promotes tumor cell invasiveness in several cancers [47]. *MYO6* upregulation drives metastasis and progression in breast, gastric, and prostate cancer [36,37,48]. The prognostic value of *KRT14*, *TRPM8*, and *MYO6* in the present study suggests they may be indicative of CTC metastatic potential (Appendix A).

CTC transcriptomes offer temporal indications for therapy resistance and metastatic potential. Label-free CTC enrichment coupled with tailored transcriptome profiling is a promising avenue for the stratification of therapy response in mCRPC. Our approach did not permit the quantification of CTC counts. Hence, the inability to establish a correlation between CTC burden and transcriptome profiles is a limitation in this study.

## 4. Materials and Methods

### 4.1. Gene Panel Development

An in-house microarray transcriptome dataset containing PCa, and leukocyte samples was used to select a panel of 64 genes [33] (Appendix A). The panel consists of genes that are highly expressed in localized PCa and mCRPC, with limited or no expression in peripheral blood cells, and/or have known roles in PCa biology. A leukocyte-specific gene, *PTPRC*, was included to assess the leukocyte background signal. *PTPRC* expression was excluded from survival analyses. The 64-gene panel will be referred to as the ‘complete gene panel’ hereafter. Mock CTC samples containing representative fractions of PCa cells and leukocytes (i.e., 10–1000 LNCaP cells spiked in 8 mL whole blood of young and healthy volunteers) were used to test the feasibility of our approach (ATCC, Manassas, VA, USA). Tumor-specificity of the complete gene panel was tested using healthy volunteer blood from age-matched males, young adult males, and age-agnostic females.

### 4.2. Patient Recruitment

Patients were recruited from a single-center observational study at Radboudumc, Nijmegen, the Netherlands (PROMPT, NCT04746300). Eligible patients were men with confirmed mCRPC. Patients were either therapy naïve or had received a maximum of one line of systemic therapy in the castrate setting. Treatment with up to 6 cycles of docetaxel or ARSI in castrate-sensitive disease was allowed. All patients provided written consent for their participation in the PROMPT study and the biobanking of blood and urine samples for pre-defined biomarker studies.

### 4.3. Healthy Donor Recruitment

EDTA blood samples from healthy volunteers were collected at the Sanquin blood bank in Nijmegen, Netherlands. In total, 29 donors participated, including 9 age-matched males (≥50 years), 10 young adult males (<50 years), and 10 females. Healthy donor demographics are shown in the Appendix A (Appendix A).

### 4.4. Next-Generation DNA Sequencing

Molecular profiling of tumor material via next-generation DNA sequencing using the TruSightOncology 500 (TSO500; Illumina, San Diego, CA, USA) panel was done for all PROMPT participants (Illumina, San Diego, CA, USA). Formalin-fixed paraffin-embedded prostate or metastatic tissue biopsy material was used for sequencing, preferably obtained in castrate state. A virtual PCa-specific diagnostic panel was applied, limiting the analysis to 44 genes with prognostic and/or druggable potential in PCa (Appendix A).

### 4.5. Presentation of Genomic Alterations

Genomic alterations from solid tissue biopsies and matched transcriptomic alterations from liquid CTC biopsies were presented in an oncoprint using the online OncoPrinter tool from cBioPortal [49,50].

### 4.6. Sample Processing

Blood was drawn into 8 mL EDTA vacutainers following inclusion and processed within 48 h. When necessary, whole blood was stored at 4 °C until processing. Blood was passed through a 6.5 µm filtration cassette at 50 mbar using the Parsortix PR1 system, according to the manufacturer’s instructions (ANGLE plc, Guildford, UK). After filtration, a pulsating backflush using phosphate-buffered saline (PBS) was passed through the cassette to harvest trapped cells in HyCEAD lysis buffer per the manufacturer’s protocol. Lysates were stored at −80 °C. All samples were shipped on dry ice to ANGLE Biosciences, Inc. laboratories in Toronto, ON, Canada for transcriptome profiling.

### 4.7. Transcriptome Profiling

Lysed samples were analyzed using sense-strand HyCEAD probes and Ziplex^®^ Flow-Thru Chip^®^ (ANGLE plc, Guildford, UK) [14,15,51]. Raw expression values were floored to 1 and log2 transformed before analysis. Transcriptome profiles were generated using unsupervised hierarchical clustering as described in Section 4.9.

### 4.8. Clinical Follow-Up and Data Collection

Patient inclusion started in September 2020 and ended in January 2021. Baseline clinical parameters and genomic alterations in metastatic tissue or prostate biopsies were recorded at the time of CTC collection. Clinical and genomic data were prospectively collected, and clinical outcomes were updated following inclusion until May 2022 or death.

### 4.9. Statistical Analysis

Clinical trajectories were visualized in a swimmer plot created in Rstudio using the following packages: Tidyverse and swimplot (Rstudio version 4.2.2) [52].

Unsupervised hierarchical clustering of log2 transformed expression values was done using Rstudio and the following packages: Tidyverse, pheatmap, pvclust, RColorBrewer, and Grid [52]. Clustering of CTC transcriptome profiles was done using Euclidean distance and complete clustering. Multiscale bootstrap resampling was done to determine the statistical significance for each cluster using 10^4^ bootstrap replications. The prognostic value of CTC gene expression, serum PSA levels, and age were determined via survival analyses using the median as a cut-off value.

Univariate and multivariate survival analysis was done in RStudio using the following packages: Tidyverse, survivalAnalysis, ggpubr, and ggstatsplot [52,53].

## 5. Conclusions

Label-free CTC enrichment provides a noninvasive and non-biased opportunity for transcriptome profiling of tumor material. CTC transcriptome profiles can stratify mCRPC patient survival and therapy response. Conventional methods for detecting therapy resistance and disease progression in mCRPC are limited. Tailored transcriptome profiling of liquid CTC biopsies is a promising avenue for the development of novel prognostic and predictive clinical tests.

## Figures and Tables

**Figure 1 ijms-24-09002-f001:**
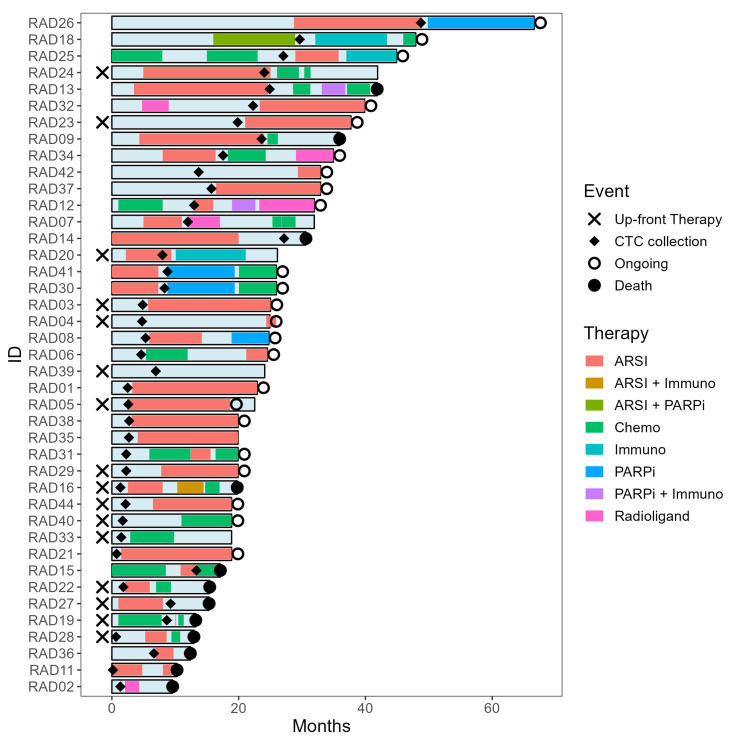
The clinical trajectories of study participants. Patient IDs were anonymized and labeled RAD (Radboudumc) accompanied by a chronological number. Event legend: indicating whether patients received prior therapy for castrate-sensitive disease (Up-front Therapy), when CTCs were collected (CTC collection), whether therapy continued (ongoing), or if a patient had become deceased (death) at the last follow-up. Therapy legend: indicating lines of monotherapy or combination therapy.

**Figure 2 ijms-24-09002-f002:**
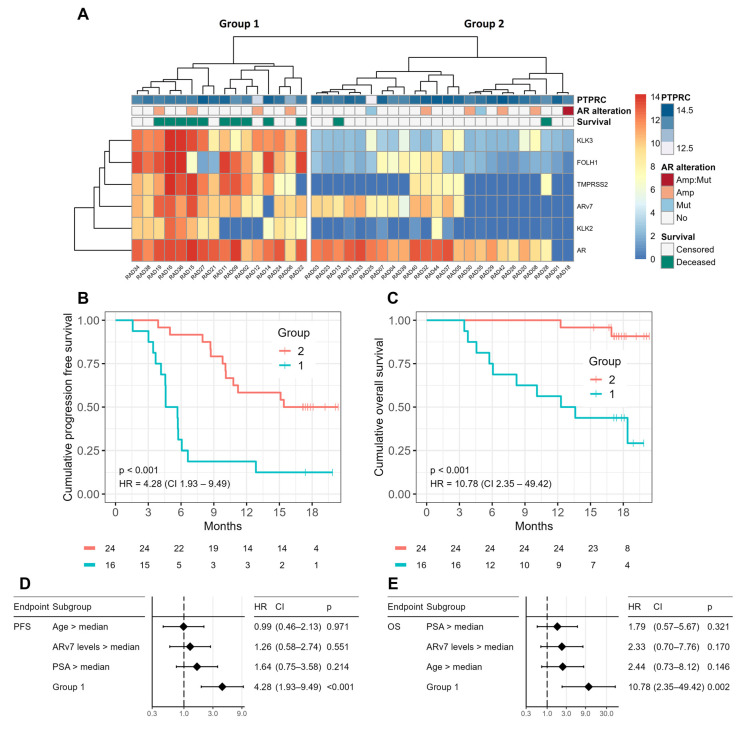
Hierarchical clustering of the therapy-agnostic cohort using the agnostic gene panel (**A**). Cluster *p*-values are *p* = 0.06 for ‘Group 1’ and *p* = 0.07 for ‘Group 2’. PFS (**B**) and OS (**C**) of ‘Group 1’ and ‘Group 2’ patients visualized in Kaplan–Meier plots. Independent prognostic values of age, *ARv7* expression, ‘Group 1’, and PSA on PFS (**D**). Independent prognostic values of age, ‘Group 1’, and PSA on PFS (**E**). Expression annotation: expression is shown in log2 values, from 0 (dark blue) to 14 (deep red). Heatmap annotation: PTPRC, leukocyte background signal in each sample; AR alteration, Amp:Mut, AR amplification and mutation, Amp, AR amplification, Mut, AR mutation, No, no alteration; Survival, Censored, alive at last follow-up, Deceased, deceased at last follow-up. Survival statistics: HR, hazard ratio; CI, 95% confidence interval; *p*, *p*-value.

**Figure 3 ijms-24-09002-f003:**
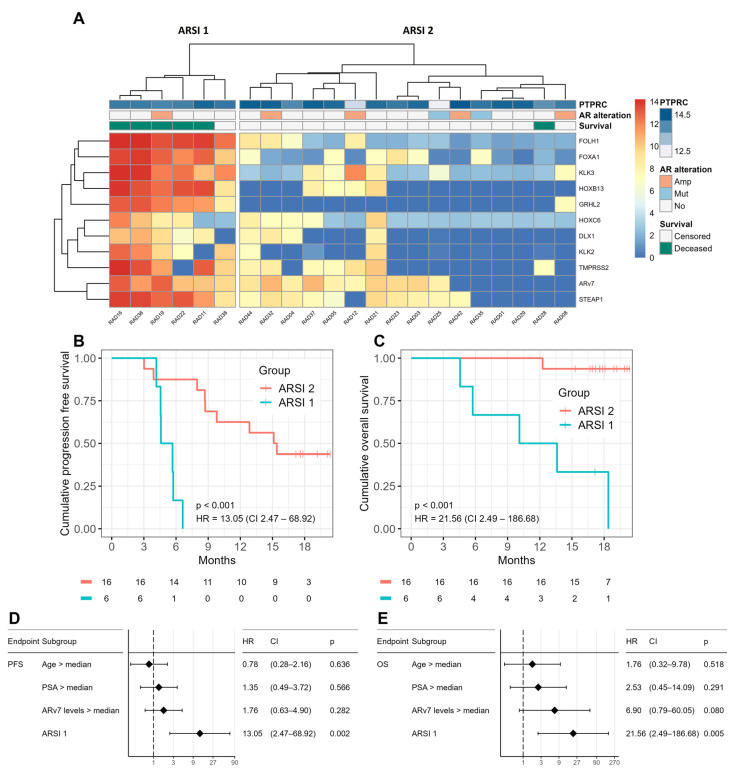
Hierarchical clustering of the ARSI cohort using the ARSI gene panel (**A**). Cluster *p*-values are *p* = 0.12 for ‘ARSI 1’ and *p* = 0.14 for ‘ARSI 2’. PFS (**B**) and OS (**C**) of ‘ARSI 1’ and ‘ARSI 2’ patients visualized in Kaplan–Meier plots. Independent prognostic values of age, PSA, *ARv7* expression, and ‘ARSI 1’ on PFS (**D**). Independent prognostic values of age, PSA, *ARv7* expression, and ‘ARSI 1’ on OS (**E**). Expression annotation: expression is shown in log2 values, from 0 (dark blue) to 14 (deep red). Heatmap annotations: PTPRC, leukocyte background signal in each sample; AR alteration, Amp, AR amplification, Mut, AR mutation, No, no alteration; Survival, Censored, alive at last follow-up (white), Deceased, deceased at last follow-up survey (green). Survival statistics: HR, hazard ratio; CI, 95% confidence interval; *p*, *p*-value.

**Figure 4 ijms-24-09002-f004:**
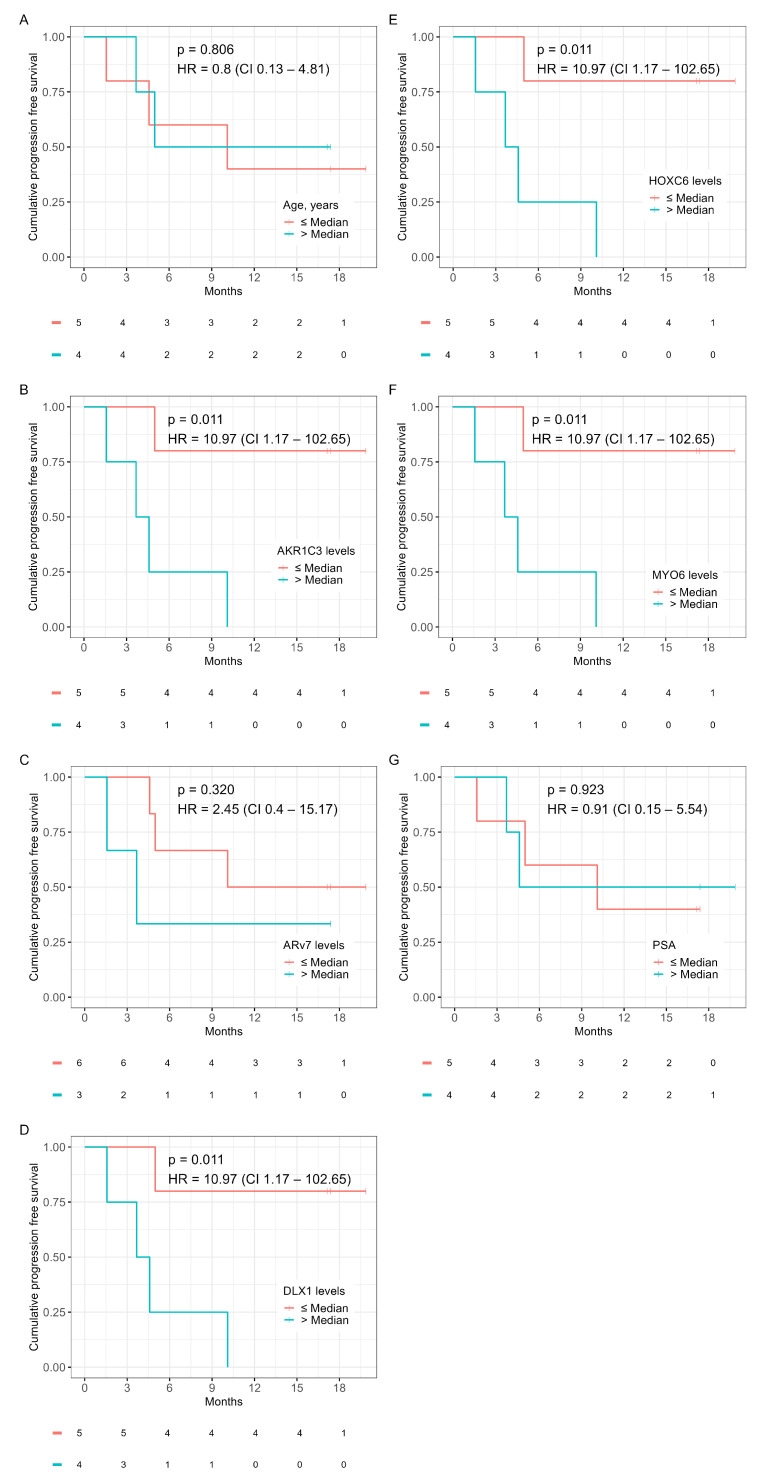
Kaplan–Meier plots for PFS in the chemotherapy cohort for age (**A**), *AKR1C3* levels (**B**), *ARv7* levels (**C**), *DLX1* levels (**D**), *HOXC6* levels (**E**), *MYO6* levels (**F**), and PSA (**G**). Survival statistics: HR, hazard ratio; CI, 95% confidence interval; *p*, *p*-value.

**Figure 5 ijms-24-09002-f005:**
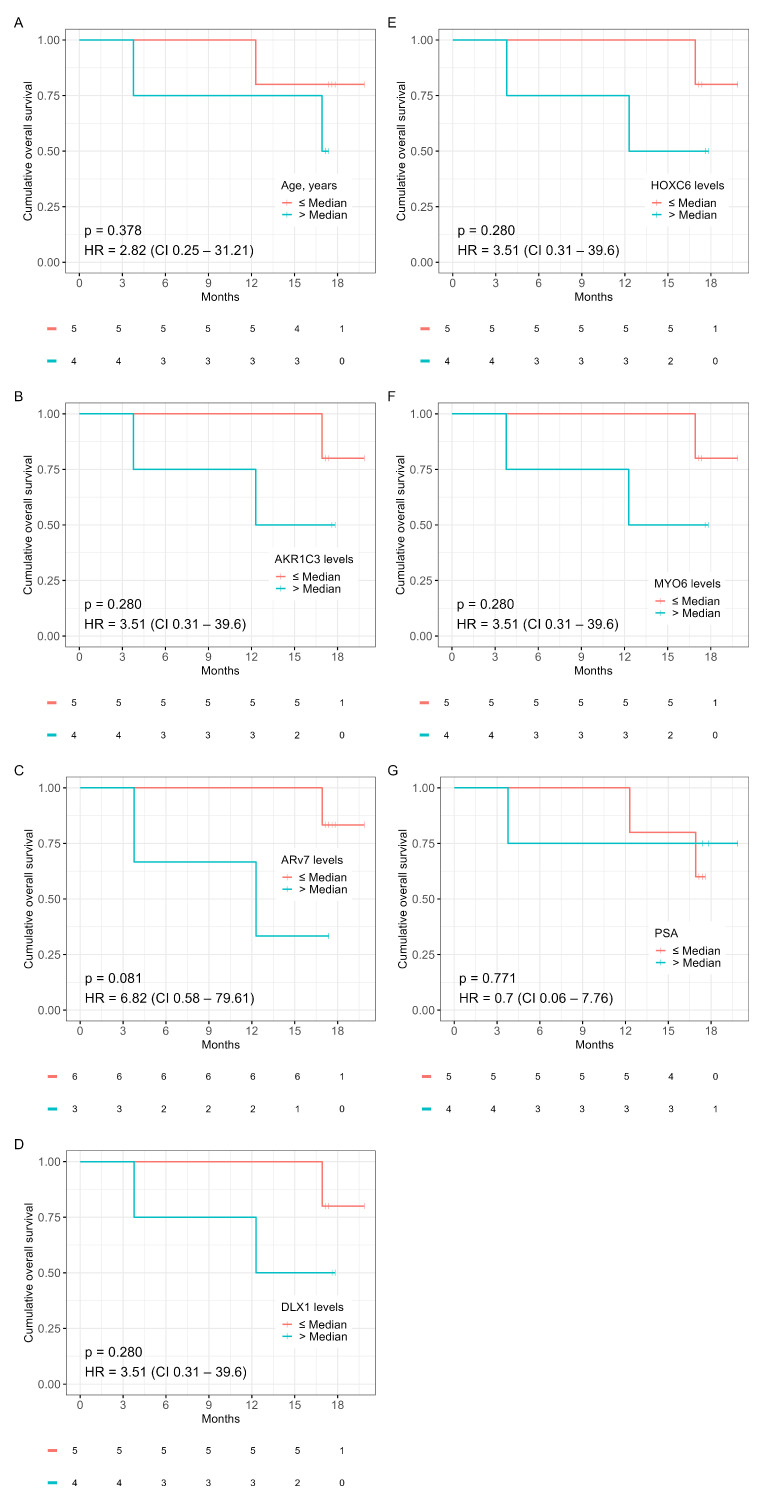
Kaplan–Meier plots for OS in the chemotherapy cohort for age (**A**), *AKR1C3* levels (**B**), *ARv7* levels (**C**), *DLX1* levels (**D**), *HOXC6* levels (**E**), *MYO6* levels (**F**), and PSA (**G**). Survival statistics: HR, hazard ratio; CI, 95% confidence interval; *p*, *p*-value.

**Table 1 ijms-24-09002-t001:** Patient characteristics.

**Number of patients**	40
Age, median (range)	65 (45–85)
Years since CRPC, the median (range)	2 (0.8–5.5)
PSA (μg/L) at inclusion, median (range)	31.5 (1.2–1600)
**De novo metastatic diseases, Num. (%)**	
Yes	29 (72.5%)
No	11 (27.5%)
**Alkaline phosphatase (U/L), median (range)**	106 (54–1866)
**Type of therapy following inclusion, Num. (%)**	
ARSI	22 (54%)
Chemotherapy	9 (22%)
Immunotherapy	2 (5%)
Radioligand therapy	2 (5%)
PARPi therapy	2 (5%)
Active surveillance	3 (7%)
**1-year survival per therapy, percentage (95% CI)**	
Therapy-agnostic	82% (72–95%)
ARSI	86% (73–100%)
Chemotherapy	89% (71–100%)
Immunotherapy	50% (13–100%)
Radioligand	50% (13–100%)
PARPi	100%
Active surveillance	33% (6.7–100%)

Number of study participants, participants’ age, years since CRPC diagnosis, median PSA level at the time of CTC collection, de novo metastasis at the time of PCa diagnosis, median alkaline phosphatase level at the time of CTC collection, type of therapy received following CTC collection, 1-year survival for each therapy group following CTC collection.

## Data Availability

Data available on request due to privacy restrictions. The data presented in this study are available on request from the corresponding author.

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
