# Peer review of "Transcriptome Profiling of Circulating Tumor Cells to Predict Clinical Outcomes in Metastatic Castration-Resistant Prostate Cancer"

_ijms, 2023, doi:10.3390/ijms24109002_

Round 1

Reviewer 1 Report

Gist/Summary: The authors perform and check CTC transcriptome profiles to stratify mCRPC patients for survival and therapy response. while they compare the conventional methods for detecting therapy resistance and disease progression, they put in use vivid use of tailored transcriptome profiling of liquid CTC biopsies  with a 64 gene panel for better prognosis. Finally, statistics and heirarcical clustering was employed to check these.

A recent article discusses the same on the sidelines of this: https://aacrjournals.org/clincancerres/article-abstract/doi/10.1158/1078-0432.CCR-22-3233/724976/PSMA-positive-Circulating-Tumor-Cell-Detection-and

Could this in analogy be used?

The manuscript is well written and justified and well taken. 

The PSMA on CTCs in men with mCRPC. could ther ebe a direct corelation  on the sidelines of who are direct beneficiaries and and who may not be by PSMA-targeting radioligand therapy. This discussion may bring better purview of teh current writeup

The population stratification and raison d'e tre was well taken but in the tailored panel for AR, could ther eby any noncoding RNAs that may be influencial in checking the activity/prognosis

In recruiting patients, were there any PSA levels recorded?

Was there any inclusion/noninclusion of prostatitis like infection in any of the validation/test cohort?

To check the heterogeneity, wouldn't hav ebeen better to have transcriptome profiling done from tumors instead of blood?  Pl justify 

Imagine a common venn between ARSI gene panel ( 64 genes) from yorus) and the otehr existing oncomine panels fo PCa, how many genes were commonly attributed to mCRPC and again any ncRNA capture?

There could be a pictorial methodology 

The P-value heuristics for clustering may be included

Scores on a scale of 0-5 with 5 being the best 

Language: 4

Novelty: 4.5

Brevity: 3.5

Scope and relevance: 4

NA

Author Response

Dear Reviewer,

Thank you for your well thought out review of our work. I hope our response satisfies your comments.

Sincerely,

Groen et al.,

Reviewer 2 Report

Groen et al. investigated the transcriptional profiling and analyzed its impact on therapy response by obtaining CTCs from Pca patients. This study uncovers a new aspect of CTCs in Pca progression. It is interesting that AR signaling pathway or newly discovered genes could be a potential biomarker for predicting therapy responses and prognosis of patients. However, to publish in the journal, I think several points should be clarified.

1)   (Page 5) Figure 2D is not referred to in the description of Figure 2. If unnecessary, this panel should be removed.

2)   (Figure 2A) The authors described “AR and BRCA2 alterations in solid tissue biopsies were associated with expression in CTCs (Page 4, Ln 112).” It would be helpful to indicate which cases of tumors have AR genomic amplification or mutation in Figure 2A. Full spell for “PTPRC” is not shown in the legends.

3)   (Page 9, Ln. 233) “64 genes were selected”: Description about these 64 genes is lacking in the Results section.

4)   (Page 9, Ln. 237)”Sperger study”: This study seems to be important for this analysis. However, the journal name is not shown in the reference 22.

5)   (Page 9, Ln. 266) They identified “AKR1C3, DLX1, HOXC6, and MYO6 were the strongest prognosticators”. However, these data are shown in supplementary figures. I think these results are important and should be moved to main figures.

Author Response

Dear Reviewer,

Thank you for your well-thought-out review of our work. I hope our response satisfies your comments.

Sincerely,

Groen et al.,
